# Dynamic Gesture Recognition Based on MEMP Network

**Xinyu Zhang** *  **and Xiaoqiang Li**

School of Computer Engineering and Science, Shanghai University, Shanghai 200444, China; xqli@i.shu.edu.cn
* Correspondence: 646599367@shu.edu.cn

**Abstract:** In recent years, gesture recognition has been used in many fields, such as games, robotics and sign language recognition. Human computer interaction (HCI) has been significantly improved by the development of gesture recognition, and now gesture recognition in video is an important research direction. Because each kind of neural network structure has its limitation, we proposed a neural network with alternate fusion of 3D CNN and ConvLSTM, which we called the Multiple extraction and Multiple prediction (MEMP) network. The main feature of the MEMP network is to extract and predict the temporal and spatial feature information of gesture video multiple times, which enables us to obtain a high accuracy rate. In the experimental part, three data sets (LSA64, SKIG and Chalearn 2016) are used to verify the performance of network. Our approach achieved high accuracy on those data sets. In the LSA64, the network achieved an identification rate of 99.063%. In SKIG, this network obtained the recognition rates of 97.01% and 99.02% in the RGB part and the rgb-depth part. In Chalearn 2016, the network achieved 74.57% and 78.85% recognition rates in RGB part and rgb-depth part respectively.

**Keywords:** gesture recognition; human computer interaction; alternative fusion neural network

---

## 1. Introduction

Gesture communication is a widely used method in people's daily lives. Gesture interaction can be used in many kinds of scenes and has rich expressive power. For instance, sign language recognition is an important application of gestures, especially in the communication between deaf and dumb people [1]. People presently pay more and more attention to the efficiency of gesture recognition. The difficulty of gesture recognition mainly lies in the difference in body shape, video background and video noise, etc. [2]. The research of gesture recognition mainly includes static aspects and dynamic aspects. Accuracy is an important criterion to measure gesture recognition algorithms. Effective gesture recognition is still a very challenging problem [3], partly due to the cultural differences, various observation conditions, noises, relative small size of fingers in images, out-of-vocabulary motions, etc.

In traditional machine learning algorithms, HMM (Hidden Markov Model) and SVM (Support Vector Machine) are often used for the recognition of gesture [4]. SVM is often paired with HOG (Histogram of Oriented Gradient) to realize the static gesture recognition of images, which is not a suitable method for dynamic gesture recognition. In the study of dynamic gesture recognition, the HMM model is applicable to the prediction of time series, and has a good recognition rate for gestures with high complexity. However, since the HMM requires more samples to complete the graph optimization, the process of training is relatively complicated. CRF (Conditional Random Field) can be trained to recognize non-classified gesture trajectories and achieve good results [5]. However, CRF training is costly and complex, and it is not suitable for large data sets.

With the rapid development of the neural network, the recognition method has gradually transferred from traditional machine learning to deep learning. In the deep learning gesture recognition



algorithm, 2D CNN and denoising self-encoder (SDAE) are often applied to feature extraction of images [6]. The 2D CNN can achieve the effect of predicting video by extracting spatial feature information of successive sets of frames in the video. 3D CNN extracts the spatio-temporal features of the entire video to get more comprehensive information [7]. Because some gesture videos may take longer time, many researchers use long short-term memory (LSTM) network to predict the gesture video [8]. LSTM can extract the timing features between frames more effectively [9]. ConvLSTM is a network structure proposed according to convolution operation and LSTM. It does not just extract spatial features like CNN, but also model according to time series like LSTM [10].

Neural networks are often used in combination when recognizing the video data set of gestures. In the combination of CNN and LSTM, videos are firstly divided into a set of frames with fixed length. Then several convolution operations and pooling operations should be carried out for each frame, and finally several feature graphs obtained after processing are predicted as input of LSTM [11]. When 3D CNN is combined with LSTM to classify video, videos should be divided into several picture sets of frames firstly, and then these frames were operated by 3D CNN in time and space according to the convolution kernel of a certain size. Finally, the processed feature set was combined to predict the input of LSTM, and we can get the accuracy [12]. However, in the combination of conventional CNN network and RNN network, single RNN operation may not obtain more accurate prediction information. Therefore, in this paper, we proposed a multi-prediction neural network with multiple mixing of 3D CNN operation and convLSTM operation. We call it the MEMP network. The MEMP network can improve the accuracy of gesture recognition. Experiments show that this network is suitable for medium and large data sets.

In the MEMP network, each gesture video needs to be split into 16 consecutive frames. These frames are subject to three consecutive 3D CNN operations and ConvLSTM. 3D CNN is used to extract spatial-temporal features of the frame set. ConvLSTM does not just predict the frame set, but also extract more spatial feature information during the prediction process. Therefore, compared with the traditional combined neural network, MEMP network retains more spatial-temporal feature information through multiple information extraction and prediction of feature maps. In dynamic gesture video, information is contained in the spatial-temporal sequence of the video. Thus, more gesture recognition accurate prediction results can be obtained. Figure 1 shows the overall structure of the network.

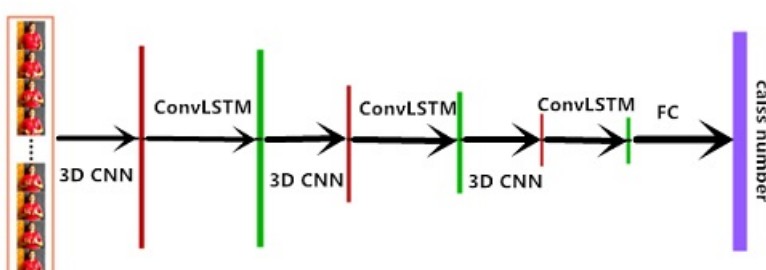

**Figure 1.** The network proposed in Figure 1. The frame set of video needs to go through three consecutive 3D CNN operations and ConvLSTM operations, and then we can get feature maps with a large amount of space-time information. Finally, feature maps are fully connected to the corresponding classification results. Each vertical line in the figure represents the state of the feature map set after a 3D CNN operation or ConvLSTM operation. The parameter transformation of each layer of the feature map is shown in Figure 2.

In the related work section, we will introduce the development of gesture recognition from traditional machine learning to deep learning. In the proposed method section, we will analyze the

internal structure of the MEMP network in detail. In the experimental part, we will use three data sets to verify the characteristics of the MEMP network.

## 2. Related Work

In the study of computer vision, getting the spatial-temporal information in videos is paramount. In traditional machine learning, Ahmed and Aly used LBP and PCA for feature extraction, HMM was also used for classification [13], they have got outstanding results. However, LBP cannot distinguish between neighborhood pixels and central pixels, or neighborhood pixels are larger than central pixels. The situation may result in the loss of information extraction. Under the work of Chen and Luo, a realtime Kinect-based dynamic hand gesture recognition system which contains hand tracking, data processing, model training and gesture classification is proposed [14]. Support Vector Machine is used as the recognition algorithm in the proposed system. Methods based on the state-of-the-art handcrafted features are difficult to handle large data sets [15]. At the same time, deep neural networks have achieved remarkable results in processing large-scale data sets [15].

Neural network structure plays an important role in the study of static gesture recognition. Xing and Li proposed a CNN structure for vision-based static hand gesture recognition with competitive performance [16]. In the dynamic gesture recognition based on deep learning, the first thing to do is preprocess the gesture video. Each video is divided into a set of frames of fixed width, length and height (the frame set in this paper is $64 \times 64 \times 16$, where the frame has a length and a height of 64 and a width of 16). RNN is often used to process those frames. Chai and Liu presented an effective spotting-recognition framework based on RNN for large-scale continuous gesture recognition [17]. Naguri and Bunescu presented a gesture recognition system based on LSTM networks, the architecture was trained on raw input sequences of 3D hand positions [18]. In many new studies of gesture recognition, people tend to use 3D CNN to extract the temporal and spatial features of video [19,20]. Zhu and Zhang used the combined network of 3dcnn and convLSTM to extract the features of video, and finally conducted classification operations through SPP (Spatial Pyramid Pooling) and FC (Fully Connected Layer) [3]. However, in traditional combined neural networks, a single RNN operation may not fully grasp the spatial-temporal information. Therefore, the MEMP network obtains more decisive information by cross-using CNN and RNN.

Although 2D CNN extracts spatial features well, 3D CNN can extract more information (spatial-temporal feature information) [7]. LSTM inherits the characteristics of most RNN models and solves the problem of gradient disappearance caused by gradual reduction of gradient backpropagation process, which is often used to predict the time characteristics of sequences [9]. However, LSTM often neglects the extraction of spatial features of feature maps in the prediction of sequences, which may affect the final spatial and temporal feature information. ConvLSTM not only has the capability of LSTM time series modeling, but also can describe local features like CNN. During the work of MEMP Neural Networks, 3D CNN is used to extract the spatial and temporal feature information of each frame, and then ConvLSTM was used to predict the set of features. Repeated 3D CNN operation and ConvLSTM operation will get more distinct information. The output size is determined by the number of gestures in the data set. ReLU is the activation function of the hidden layer, and it can improve the learning speed of neural networks at different depths. This means that using ReLU activation function can avoid the vanishing gradient problem [21].

## 3. Proposed Method

In this section, we will describe the internal structure of the MEMP Neural Networks in detail. Predicted results were obtained after the video frame sets operated by three consecutive 3DCNN operations and ConvLSTM operations.

CNN plays a significant role in image feature extraction [22]. Here F = $\{f_1, f_2, .....f_{16}\}$ represents the frame set of video. The size of $f_i$ is $64 \times 64$. After a 3D CNN operation, the size of each feature map was not changed, and the number of channels increased to 8. This operation can extract

spatial-temporal information of feature maps. Then, after a convLSTM operation, the set became F′ = $\left\{ f'_{1,1}, f'_{1,2}, \ldots f'_{16,16} \right\}$. The number of channels increased to 16, the number of each channel was 16 and the size of feature map is 64 × 64. Then, through a 3D POOL operation, the number of feature maps in each channel and the length and width of each feature map were reduced by half. After similar operations of 3DCNN, convLSTM and POOL, the size of feature set became F″ = $\left\{ f''_{1,1}, f''_{1,2}, \ldots f''_{64,4} \right\}$, with 64 channels. The number of each channel was 4, and the size of feature map was 16 × 16. After above operations, the spatial-temporal feature information of the frame set will be retained. At the end of the full connection layer, the results will be classified to obtain different types of gestures. The specific operation is as follows.

3D CNN is good at extracting features of video [23]. So in MEMP Neural Networks, 3D CNN is used to process spatial and temporal features of feature graph sets. As shown in Figure 2, there are three 3D convolution operations (C1, C2, C3), three convLSTM operations (CL1, CL2, CL3) and two pooling operations (P1, P2) in the network. The size of the three 3D convolution kernels is 2 × 2 × 2, and the step length is 1 × 1 × 1. Since the convolution mode of 'SAME' is adopted, the size of each feature map in the convolution operation will not change. The filter sizes corresponding to the three convolution operations are 8, 32 and 64 respectively. The pooled method is 'max pooling' and the pooled size is 2 × 2 × 2. After each cisterization operation, a 'droupout' layer will be followed. 'droupout' will effectively reduce the occurrence of over-fitting, which can reach the effect of regularization to a certain extent [24]. The formula of 3D CNN is as follows:

$$v_{ij}^{xyz} = ReLU(b_{ij} + \sum_{m} \sum_{p=0}^{P_i-1} \sum_{q=0}^{Q_i-1} \sum_{r=0}^{R_i-1} w_{ijm}^{pqr} v_{(i-1)m}^{(x+p)(y+q)(z+r)}) \tag{1}$$

'*ReLU*' is the activation function of the hidden layer. $v_{ij}^{xyz}$ represents the current value of the coordinates (x, y, z) in the *i*-th and *j*-th feature graph sets. $b_{ij}$ represents the bias of the *i*-th layer and the *j*-th feature graph set, $w_{ijm}^{pqr}$ represents the weight of the *m*-th filter connected by the position (p, q, r) in the *i*-th layer and *j*-th feature graph set. $P_i$, $Q_i$ and $R_i$ represent the height, width and depth of the convolution kernel respectively [23].

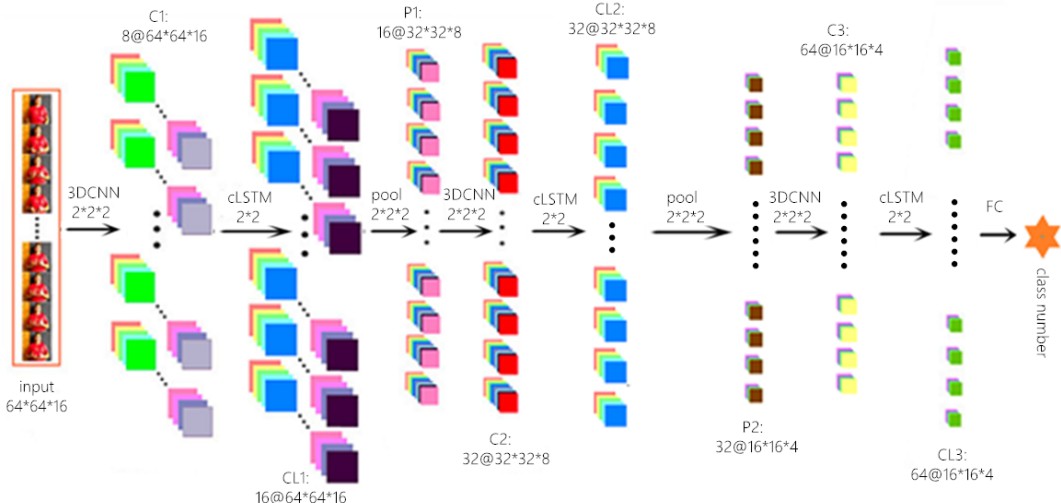

**Figure 2.** In Figure 2, the MEMP Neural Networks structure as well as the change of the feature graph in each layer of the network structure are shown. The whole process is divided into three parts, video processing, (3D CNN-convLSTM)*3 and FC. Initially, each video is split into 16 consecutive frames. Secondly, these feature sets need to be performed three convolution operations, three convLSTM operations and two pooling operations to obtain more spatial-temporal information. Finally, the network connects the full connection layer to classify these results.

Compared with traditional LSTM, convLSTM can better extract spatial and temporal features of feature graph sets [10]. The reason is that ConvLSTM can consider the spatial information of a single feature map when it processes and predicts time series events. So ConvLSTM can solve timing problems in dynamic gesture recognition more effectively. In this experiment, the processed feature graph set need to be further extracted by convLSTM after 3D CNN operation. The convolution kernel is $2 \times 2$, and the step lengths is $1 \times 1$. The convolution way is 'SAME', so the size of the feature graph is not changed. The final output filter size is 64. The main formula of convLSTM is as follows:

$$i_t = \sigma(W_{xi} * X_t + W_{hi} * H_{t-1} + W_{ci} \circ C_{t-1} + b_i) \tag{2}$$

$$f_t = \sigma(W_{xf} * X_t + W_{hf} * H_{t-1} + W_{cf} \circ C_{t-1} + b_f) \tag{3}$$

$$C_t = f_t \circ C_{t-1} + i_t \circ \tanh(W_{hc} * H_{t-1} + W_{xc} * X_t + b_c) \tag{4}$$

$$o_t = \sigma(W_{xo} * X_t + W_{ho} * H_{t-1} + W_{co} \circ C_t + b_o) \tag{5}$$

$$H_t = o_t \circ \tanh(C_t) \tag{6}$$

where $X_1 \ldots X_t$ are input, $C_1 \ldots C_t$ are unit output, and $H_1 \ldots H_t$ are hidden layer. $i_t$, $f_t$ and $o_t$ are three dimensional tensor of convLSTM respectively. The last two dimensions are spatial dimension (row and column). '$\circ$' is the convolution operation, and '*' is 'Hadamard product' [10]. The feature set passes through a full connection layer after a ConvLSTM operation, and the output size of the full connection layer is based on the number of gestures. The final optimization function adopts the 'Adam' algorithm. 'Adam' has high computational efficiency and low memory demand and has no impact on the gradual diagonal scaling. So it is very suitable for the problem of large data processing [24].

## 4. Experiment

### 4.1. Datasets

This experiment used three dynamic gesture data sets: LSA (Argentinian Sign Language), chalearn 2016 (IsoGD) and SKIG.

LSA: This data set represents the Argentine sign language. The LSA data set include 3200 RGB videos and 10 non-expert subjects repeated 64 different LSA signs five times. Each video has a resolution of $1920 \times 1080$, 60 frames per second [25]. Table 1 shows the statistical information of training part and test part in LSA dataset.

**Table 1.** Statistical information of LSA dataset.

|          | Gesture Video |
| -------- | ------------- |
| All      | 3200          |
| Training | 80% randomly  |
| Testing  | 20% randomly  |

IsoGD: This is a large-scale gesture data set, which is derived from chalearn gesture data set [26]. The data set contains 47,933 rgb-depth video gestures made by a total of 249 gestures made by 21 different individuals. Table 2 shows the statistical information of training part and test part in IsoGD dataset.

**Table 2.** Statistical information of IsoGD dataset.

|  | Gesture Video | RGB Part | Depth Part |
|---|---|---|---|
| Training | 35,878 | 35,878 | 35,878 |
| Validation | 5784 | 5784 | 5784 |
| Testing | 6271 | 6271 | 6271 |

SKIG: This data set contains 1080 rgb-depth gesture sequences collected from six individuals. All of these sequences were shot synchronously through the Kinect sensor, which includes an RGB camera and a depth camera. A total of 10 gestures were collected in this data set [27]. Table 3 shows the statistical information of training part and test part in SKIG dataset.

**Table 3.** Statistical information of SKIG dataset.

|  | Gesture Video | RGB Part | Depth Part |
|---|---|---|---|
| All | 1080 | 1080 | 1080 |
| Training | 80% randomly | 80% randomly | 80% randomly |
| Testing | 20% randomly | 20% randomly | 20% randomly |

### 4.2. Video Processing

The dynamic gesture data set is generally composed of a large number of video. Video's resolution and time length will be different, so each video needs to be preprocessed. Videos will be split into a set of 16 consecutive frames with uniform time first, and then all frames need to be resized ($64 \times 64$).

### 4.3. Implementation

In terms of hardware environment, the graphics card used in our experiment is Quadro P5000 (NVIDIA, USA), CPU is E5 2650 v4 (Intel, USA), and the memory size is 128 G. Our development system is Windows 10 and the development tool used is PyCharm 2018.1 (JetBrains, Czech Republic). We use keras along with tensorflow. All data sets video are split into 16 consecutive frames of $64 \times 64$. Then 64 batches were processed when processing data, and all data were trained 100 times in total. The initial learning rate of the training process is 0.001. The exponential decay rate of the first order moment estimate is 0.9, and the exponential decay rate of the second moment estimate is 0.999. At last, the softmax function is used for the full connection classification, and the activation function is 'Adam'.

### 4.4. Experimental Results

**LSA**: 2D CNN, 3D CNN, 3D CNN + LSTM and the MEMP network structure were respectively used to train LSA data set. The network structure proposed in the paper [11] is 2D CNN + LSTM. Therefore, there are four groups of experiments in this data set, and the experimental results are shown in Table 4.

**Table 4.** LSA experimental results.

| Experiment Dataset | LSA | |
|---|---|---|
| Records | Method | Accuracy |
| 1 | 2D CNN | 93.563% |
| 2 | 2D CNN + LSTM [11] | 95.217% |
| 3 | 3D CNN | 97.892% |
| 4 | 3D CNN + LSTM | 98.451% |
| 5 | Our Method (MEMP network) | 99.063% |

As can be seen from Table 4, the MEMP neural network structure proposed in this paper has improved the recognition rate of LSA data sets by 3.846% compared with the structure proposed in this paper [11], and the MEMP network is 1.171% higher than the 3D CNN network frequently used in video processing. It can be seen that this network structure has high accuracy in processing LSA data sets.

Figure 3 show the accuracy and loss function change of LSA data set during the training of this network. The epoch of the entire network is 100. The loss function used is 'categorical-crossentropy'. The yellow line represents the verification set and the blue line represents the training set. From this we can see that when the epoch reaches 40, the accuracy and loss function of the network tends to be stable. This shows the stability of the network.

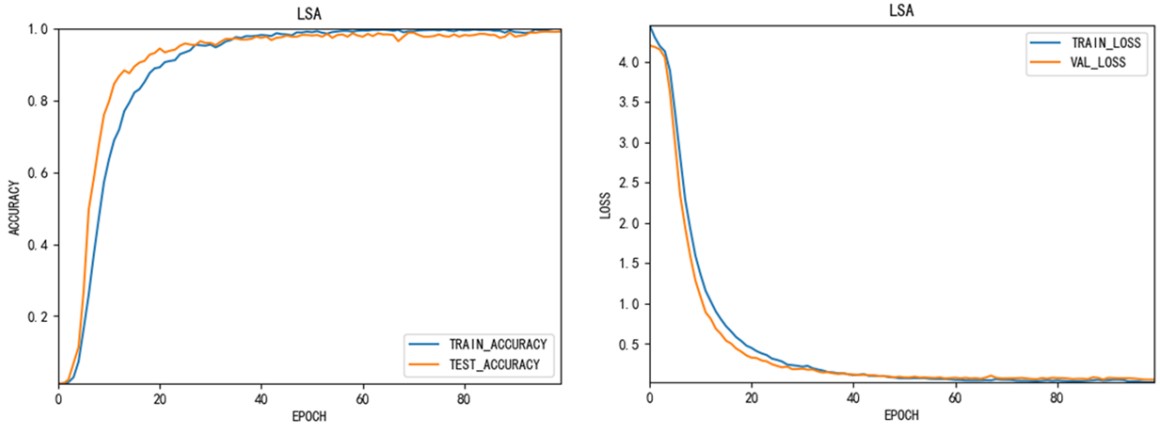

**Figure 3.** Accuracy on LSA dataset using MEMP network.

**IsoGD**: in this data set, the MEMP network and the 3D CNN network are used to train RGB and rgb-depth data sets respectively. In the paper [20], the author used C3D and LSTM networks to train IsoGD data. The experimental results are shown in Table 5.

**Table 5.** IsoGD experimental results.

| Experiment Dataset | IsoGD | |
|---|---|---|
| Records | Method | Accuracy |
| 1 | 2D CNN (RGB) | 46.03% |
| 2 | 2D CNN (RGB-Dep) | 49.17% |
| 3 | 3D CNN (RGB) | 49.68% |
| 4 | 3D CNN (RGB-Dep) | 55.12% |
| 5 | 3D CNN and ConvLSTM (RGB) [3] | 43.88% |
| 6 | 3D CNN and ConvLSTM (RGB-Dep) [3] | 51.02% |
| 7 | Res-C3D and Skeleton LSTM (RGB-Dep) [20] | 68.42% |
| 8 | ResNet-18 (RGB-Dep) | 66.18% |
| 9 | ResNet-34 (RGB-Dep) | 67.54% |
| 10 | 3DResNet-18 (RGB-Dep) | 71.24% |
| 11 | MEMP network (RGB) | 74.57% |
| 12 | MEMP network (RGB-Dep) | 78.85% |

It can be seen from the results that, in the IsoGD data set, the accuracy of MEMP network is 10.43% higher than that of the paper [20], which is a big breakthrough. In the same data set, MEMP network was 24.89% higher in RGB and 23.73% higher in rgb-depth data sets than the commonly used 3D CNN network. In comparison with the paper [3], we have improved 30.69% and 27.83% in RGB and RGB-Depth part respectively. These improvements illustrate the importance of multi-fetch multi-prediction for frame sets. In the experiment, the size of frame is 64 × 64, which is smaller

than the size of paper [3] and paper [20]. This shows that MEMP network can save a lot of time. Compared with the current popular ResNet and 3D ResNet, the MEMP network also has a significant improvement in accuracy.

Figure 4 shows the variation of accuracy and loss function in training IsoGD data set. Figure 5 shows the accuracy comparison between 2D CNN, 3D CNN and our method (MEMP network) in the IsoGD (RGB-Depth) data set. From this we can see that our method in large data sets are superior to the mainstream deep learning networks in both accuracy and speed of convergence.

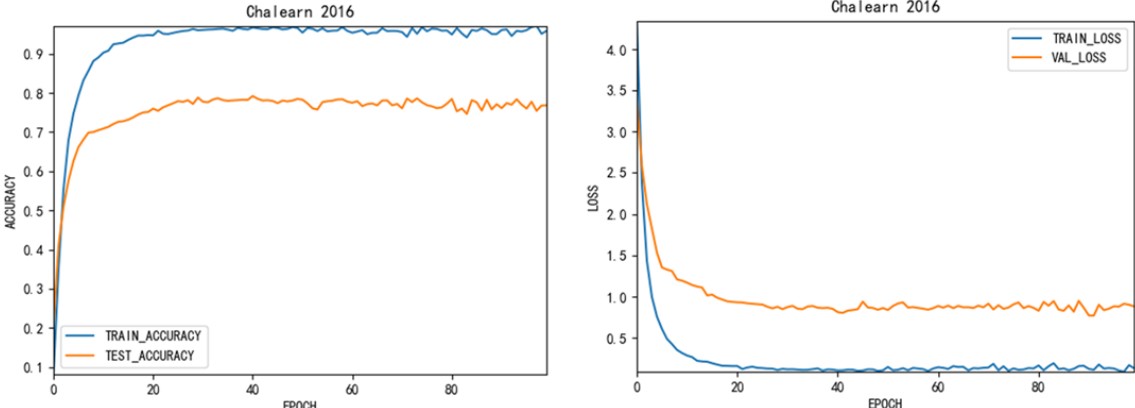

**Figure 4.** Accuracy on IsoGD dataset using MEMP network.

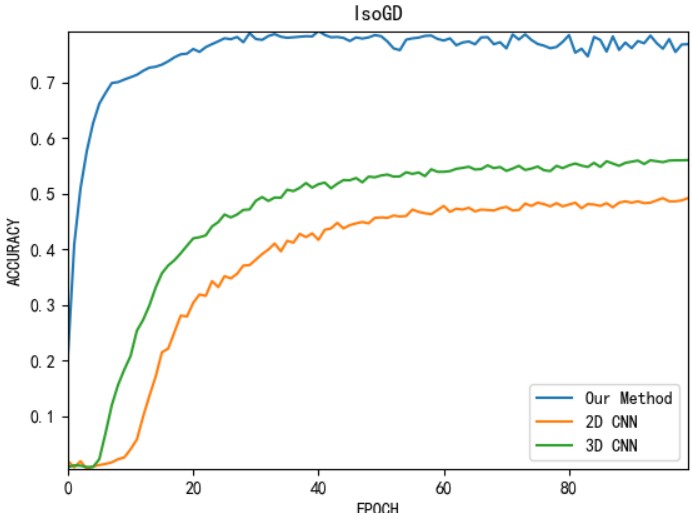

**Figure 5.** Accuracy comparison on IsoGD using mainstream method.

**SKIG**: in the paper [28], the author proposed a network structure of LPSNet to train SKIG data sets. In this experiment, the RGB data set and rgb-depth data set in SKIG were trained respectively, and the results are shown in Table 6.

**Table 6.** SKIG experimental results.

| Experiment Dataset | SKIG | |
|---|---|---|
| Records | Method | Accuracy |
| 1 | LPSNet (RGB) [28] | 96.7% |
| 2 | LPSNet (RGB-Depth) [28] | 98.7% |
| 3 | 3D CNN and ConvLSTM (RGB) [3] | 95.93% |
| 4 | 3D CNN and ConvLSTM (RGB-Dep) [3] | 98.70% |
| 5 | MEMP network (RGB) | 97.01% |
| 6 | MEMP network (RGB-Dep) | 99.02% |

As can be seen from the results in Table 6, accuracies of MEMP network in the RGB part and the rgb-depth part are higher than the LPSNet proposed in the paper [28]. Compared with the paper [3], our results are also superior to theirs.

Figure 6 shows the variation of accuracy and loss function in the training SKIG data set. As can be seen from the graph above, MEMP network has achieved outstanding results in different data sets. This indicate that our method can be applied to many medium and large data sets.

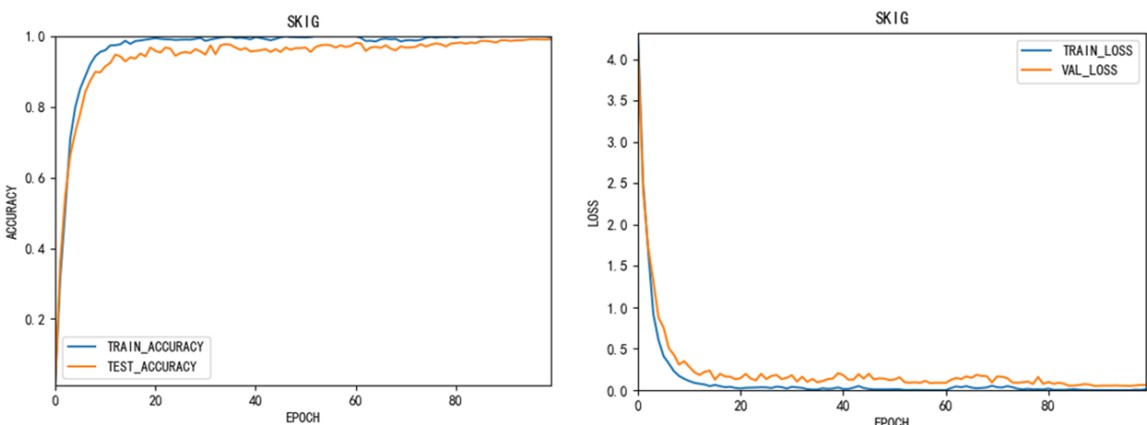

**Figure 6.** Accuracy on SKIG dataset using MEMP network.

## 5. Conclusions

Gesture recognition plays an important role both in daily life and in the direction of computer vision. The current method based on deep learning is the main research aspect of gesture recognition. In this paper, we proposed a MEMP network for gesture recognition research. The advantage of MEMP network is that 3D CNN and convLSTM are mixed several times to extract and predict the video gesture information multiple times, so as to get higher accuracy. The MEMP Neural Networks has achieved high accuracy in LSA, IsoGD and SKIG data sets, and it also indicates that this network is applicable in many medium and large video data sets. ResNet can easily realize good accuracy of image classification and location tasks. In future studies, we will use the residual network to classify gesture recognition.

**Author Contributions:** Methodology, Review, Writing and Editing, X.Z.; Validation, X.L.

**Funding:** This work is partially supported by Shanghai International Cooperation Fund Project (No. 12510708400) and Shanghai Innovation Action Plan Project (No. 16511101200) of Science and Technology Committee of Shanghai Municipality.

**Conflicts of Interest:** We have no conflict of interest.

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
