# Peer review of "Dynamic Gesture Recognition Based on MEMP Network"

_futureinternet, doi:10.3390/fi11040091_

Reviewer 1 Report

Need some grammar check

Author Response

Dear sir or madam:

Thank you for reviewing my paper and for making valuable revisions.I have modified the paper according to your comments, please refer to the modification section below.

Reviewer 2 Report

The authors proposed a new approach composed of two known deep neural networks (3D CNN and ConvLSTM) for gesture recognition. The proposed approach was verified experimentally and the results were compared to those obtained by other approaches.

The following issues should be addressed:

1. Please thoroughly analyze the pros and cons of the proposed method when compared with other state-of-the-art approaches.

2. The English language should be corrected – there are grammar and style errors. The paper should be corrected by a professional service.

Author Response

Dear sir or madam:

Thank you for reviewing my paper and for making valuable revisions. I have modified the paper according to your comments, please refer to the modification section below.

Point 1. Please thoroughly analyze the pros and cons of the proposed method when compared with other state-of-the-art approaches.

Response 1: I added the current popular ResNet and 3DResNet in the experimental part of the IsoGD dataset experiment. The residual network is suitable for large data sets. Some time ago I also used the residual network to train the LSA dataset and the SKIG dataset, but the effect was not good.

Point 2. The English language should be corrected – there are grammar and style errors. The paper should be corrected by a professional service.

Response 2: I have carefully checked the whole grammar and revised it.

Reviewer 3 Report

The paper proposes a novel model called MEMP for dynamic gesture recognition. The model is interesting and the results are convincing. Regarding the current manuscript, I have the following concerns:

1. The authors should describe the structure of the manuscript at the end of Introduction.

2. What are the main contributions of this paper? They should be listed in Introduction.

3. In Figure 1, are the red or green bars are feature vectors? The authors should explain the figure in detail.

4. The authors don't need to list the first three authors of a reference when introducing their work. Usually, listing the Last name of the first two authors is ok.

5. I suggest the authors list the statistical information of the datasets in a table.

6. What is the relationship between Figure 1 and Figure 2?

7. How did the authors split your data for training, validation, and testing? 

8. The authors say "SVM is often paired with HOG(Histogram of Oriented Gradient) to realize the static gesture recognition of images". Why not compare the proposed approach with such traditional methods?

9. Besides keras, what are the environments of hardware and software?  The authors should discuss the running time of the proposed approach.

10. What are the limitations of the proposed approach?

11. Why does the MEMP work for dynamic gesture recognition? What are its advantages?

12. Many typos or grammatical errors in the current version. The authors need to doublecheck the manuscript.

For example, in Abstract, "three data sets(LSA64, SKIG and Chalearn 2016) is..." is=>are

"Although 2D CNN extract..." extract=>extracts

"Figure 4 respectively show..." show=>shows

...

and so many others

13. BTW, I suggest that the authors provide the line number in the paper for review.

14. References should be in a uniform format. Please double-check the references.

Author Response

Dear sir or madam:

Thank you for reviewing my paper and for making valuable revisions. I have modified the paper according to your comments, please refer to the modification section below.

Point 1. The authors should describe the structure of the manuscript at the end of Introduction.

Response 1: I have added a brief introduction to the relevant work, the proposed method, and the experimental part at the end of the introduction.

Point 2. What are the main contributions of this paper? They should be listed in Introduction.

Response 2: I have already indicated in the introduction part that the MEMP network improves the accuracy of gesture recognition. And the data set that the MEMP network can fit.

Point 3. In Figure 1, are the red or green bars are feature vectors? The authors should explain the figure in detail.

Response 3: In the comments in Figure 1, we explain the meaning of each vertical line.

Point 4. The authors don't need to list the first three authors of a reference when introducing their work. Usually, listing the Last name of the first two authors is ok.

Response 4: I reduced the number of authors in the citation to two and kept their last names.

Point 5. I suggest the authors list the statistical information of the datasets in a table.

Response 5: I added three table of statistics in the experiment section.

Point 6. What is the relationship between Figure 1 and Figure 2?

Response 6: The relationship between Figure 1 and Figure 2 is added in the explanation section of Figure 1.

Point 7. How did the authors split your data for training, validation, and testing?

Response 7: The explanation of the training set validation sets and test sets for the three datasets is illustrated in the three new tables.

Point 8. The authors say "SVM is often paired with HOG(Histogram of Oriented Gradient) to realize the static gesture recognition of images". Why not compare the proposed approach with such traditional methods?

Response 8: As the traditional methods(SVM,HOG) I introduced are mainly applicable to static gesture recognition, the proposed network is composed of 3D CNN and ConvLSTM. This network structure is suitable for dynamic gesture recognition.These traditional dynamic gesture recognition methods are not suitable for use on large data sets.

Point 9. Besides keras, what are the environments of hardware and software? The authors should discuss the running time of the proposed approach.

Response 9: The hardware environment and software environment I have developed have been added in the experimental part of the paper.‘In terms of hardware environment, the graphics card used in our experiment is NVIDIA P5000, CPU is E5 2650 v4, and the memory size is 128G. Our development system is Windows 10 and the development tool used is PyCharm 2018.1.’

’

Point 10. What are the limitations of the proposed approach?

Response 10: In the introduction section we added a note that the MEMP network is suitable for medium to large data sets, which means that it is not suitable for small data sets.

Point 11. Why does the MEMP work for dynamic gesture recognition? What are its advantages?

Response 11: In the introduction section, a new explanation has been added. The information of dynamic gesture recognition is contained in the spatiotemporal sequence of video, while the characteristics of MEMP network are predicted multiple times with spatiotemporal information.

In fact, MEMP also achieved good results in behavior recognition. For example, in the UCF-101 data set, we achieved an accuracy of 87.2%.

Point 12. Many typos or grammatical errors in the current version. The authors need to doublecheck the manuscript.

Response 12: I have carefully checked the whole grammar and revised it.

Point 13. BTW, I suggest that the authors provide the line number in the paper for review.

Response 13: I have provide line number.

Point 14. References should be in a uniform format. Please double-check the references.

Response 14: I have reworked the references in the format of the template.

Round  2

Reviewer 3 Report

The author responded well to my concerns and I think it can be accepted.